# Impact of the Nanocarbon on Magnetic and Electrodynamic Properties of the Ferrite/Polymer Composites

**DOI:** 10.3390/nano12050868

**Published:** 2022-03-04

**Authors:** Alex V. Trukhanov, Daria I. Tishkevich, Svetlana V. Podgornaya, Egor Kaniukov, Moustafa A. Darwish, Tatiana I. Zubar, Andrey V. Timofeev, Ekaterina L. Trukhanova, Vladimir G. Kostishin, Sergei V. Trukhanov

**Affiliations:** 1Department of Electronic Materials Technology, National University of Science and Technology MISiS, 119049 Moscow, Russia; dashachushkova@gmail.com (D.I.T.); podgsv@mail.ru (S.V.P.); ka.egor@mail.ru (E.K.); andtim2011@gmail.com (A.V.T.); katu-shkak@mail.ru (E.L.T.); drvgkostishyn@mail.ru (V.G.K.); 2Laboratory of Magnetic Films Physics, Scientific-Practical Materials Research Centre of National Academy of Sciences of Belarus, 220072 Minsk, Belarus; fix.tatyana@gmail.com (T.I.Z.); sv_truhanov@mail.ru (S.V.T.); 3Physics Department, Faculty of Science, Tanta University, Tanta 31527, Egypt; mostafa_ph@science.tanta.edu.eg

**Keywords:** composite materials, hexaferrites, polymer matrix, nanosized carbon, magnetic and electrodynamic properties

## Abstract

Binary and ternary composites (CM) based on M-type hexaferrite (HF), polymer matrix (PVDF) and carbon nanomaterials (quasi-one-dimensional carbon nanotubes—CNT and quasi-two-dimensional carbon nanoflakes—CNF) were prepared and investigated for establishing the impact of the different nanosized carbon on magnetic and electrodynamic properties. The ratio between HF and PVDF in HF + PVDF composite was fixed (85 wt% HF and 15 wt% PVDF). The concentration of CNT and CNF in CM was fixed (5 wt% from total HF + PVDF weight). The phase composition and microstructural features were investigated using XRD and SEM, respectively. It was observed that CM contains single-phase HF, γ- and β-PVDF and carbon nanomaterials. Thus, we produced composites that consist of mixed different phases (organic insulator matrix—PDVF; functional magnetic fillers—HF and highly electroconductive additives—CNT/CNF) in the required ratio. VSM data demonstrate that the main contribution in main magnetic characteristics belongs to magnetic fillers (HF). The principal difference in magnetic and electrodynamic properties was shown for CNT- and CNF-based composites. That confirms that the shape of nanosized carbon nanomaterials impact on physical properties of the ternary composited-based magnetic fillers in polymer dielectric matrix.

## 1. Introduction

Today, the development of the functional composites based on magnetic (such as ceramic/ceramic or hard/soft composites [1,2,3] and ceramic/polymer composites [4,5]) and carbon (nanotubes, graphite and nanoplatelet) [6,7,8,9,10] nanomaterials attracts great attention due to the fundamental significance and practical importance. The fundamental significance is associated with the violation of the principle of additivity. From the practical point of view, these composites can be used for various applications in functional devices as active media [2,3,5]. These materials can be used in the field of high-frequency applications such as electromagnetic absorbers [1,2,3], antenna technology [6] and electromagnetic compatibility [10]. This can be explained by the combination and synergy of the useful properties of the polymer matrix and powdered fillers. Thus, the development of such kinds of materials is a promising direction of modern materials science, which is associated with a high potential for their use. Composite material (CM) based on polymers combine the advantages of matrix (high processability, flexibility, plasticity) and fillers (high electrical conductivity, mechanical strength, heat resistance, magnetic properties). CM allows the obtaining of new materials with properties that cannot be achieved using single-phase materials. In addition to these advantages, CM has no disadvantages of single-phase materials, such as skin-effect, ability to oxidize, weak temperature stability, and a narrow range for practical applications. Analysis of experimental data has shown that materials based on the dielectric organic matrix with dispersed fillers from multicomponent magnetic oxides (in particular ferrites) are widely used in practice [11,12,13]. There are a lot of materials that can be used as powdered fillers. As a rule, researchers focused on magnetic powders and carbon nanomaterials (nanotubes, nanoflakes and nanoclusters) as perspective fillers to control the magnetic and electrical properties of binary composites. Magnetic materials with controlled properties based on many composite oxides attract great attention due to their fundamental and practical aspects. Iron-based complex oxides or ferrites with a hexagonal structure (hexaferrites or HF) are the most important class of magnetic materials [14,15,16,17]. They are widely discussed by many researchers and are used for practical applications as magnets with high coercivity and remnant magnetization, as magnetic memory materials, and as electromagnetic absorbers of high-frequency radiation. Interest in HF is due to the strong correlation between chemical composition, structural, magnetic, and electrical parameters, and the ability to influence physical properties by changing the chemical composition. The chemical composition of HF and their local atomic structure are two main parameters determining the intensity of indirect exchange interactions, which are manifested by changing <Fe-O> bond lengths and <O-Fe-O> angles [18,19]. Bond lengths and valent angles determine magnetic and electrical parameters. One of the main approaches to improving the properties of ferrites is the chemical substitution of Fe^3+^ ions by diamagnetic and paramagnetic ions. The most interesting case for controlling magnetic, electrical and electrodynamic properties is the substitution of the Fe^3+^ by the Al^3+^ ions [20,21,22,23]. Al^3+^ ions are diamagnetic ions and have a different ionic radius and electronic configuration that results in the changes in magnetic and electrical properties of the substituted HF. Frequently, researchers obtain the required magnetic, electrical and microwave properties by preparing the HF-based composite [2,4,7]. The HF/polymer composite prepared by thermal pressing or dispersion of HF powders in polymers with high electrical permittivity is the most interesting from a practical standpoint [24,25]. Polyvinylidene fluoride, or PVDF, is obtained by polymerization of vinylidene fluoride (CH_2_=CF_2_). PVDF is a valuable semi-crystalline and thermoplastic polymer due to its high chemical stability, high flexibility, resistance to organic solvents, strength and high modulus of elasticity compared to other polymers, excellent thermal and fire resistance, as well as the presence of piezoelectric and pyroelectric properties. The latter are especially important for applications in radio and microsystem technology. In particular, PVDF has proven to be a very useful component in EMI shielding materials. Therefore, we chose it as the basis of matrix composites to overcome such shortcomings of hexaferrite as brittleness and the impossibility of giving products the desired shape. At the same time, to obtain specific functional, electrical, and mechanical characteristics, it is advisable to use not pure PVDF but its copolymers, the most common of which is a copolymer of PVDF with tetrafluoroethylene (TFE). For the preparation of composites, we used a copolymer of the F2M brand (hereinafter simply PVDF) as a polymer matrix. Here, a polymer composite based on Al-substituted M-type hexaferrite (HF) with carbon nanomaterials (carbon nanotube, or CNT, and carbon nanoflake, or CNF) as fillers has been developed. We need to note that it was the first time we demonstrated and analyzed the critical impact of the type of nanocarbon (first of all the shape and surface activity) on magnetic and electrodynamic properties.

## 2. Materials and Methods

### 2.1. Synthesis and Materials

To establish the influence of the type of carbon nanomaterials (CNT and CNF) on magnetic and electrodynamic parameters we produced ternary composites based on Al-substituted HF (BaFe_11.7_Al_0.3_O_19_) in a polymer matrix (PVDF) with the addition of multilayer CNT and CNF. We have synthesized HF by the solid state reaction method [26,27,28] from initial oxides (Fe_2_O_3_, Al_2_O_3_) and carbonate (BaCO_3_) in accordance with the reaction:(1)$${\mathrm{BaCO}}_{3}+5.85{\mathrm{Fe}}_{2}O_{3}+0.15{\mathrm{Al}}_{2}O_{3\;}\to {\;\mathrm{BaFe}}_{11.7}{\mathrm{Al}}_{0.3}O_{3}+{\mathrm{CO}}_{2}↑$$

CNT is a commercially available product produced by CheapTubes Ins (Grafton, WV, USA). CNF was produced by colleagues from Phys.-Chem. Tech. Lab. of S&P Materials Research Center [29]. The ratio between HF and PVDF in HF + PVDF composite was fixed (85 wt% HF and 15 wt% PVDF). The concentration of CNT and CNF in Al-HF/PVDF mixture was fixed (5 wt% from total HF + PVDF weight). The process of synthesis of HF + PVDF + CNT and HF + PVDF + CNF was carried out by thermal pressing of mechanically mixed initial powders of a polymer matrix, HF and carbon nanomaterials. The mixing process took place in two stages. At the first stage, powders of HF and carbon (CNT or CNF) were mixed. The mixing process consisted of the mechanical grinding of powders in an agate mortar with the addition of ethanol (for more uniform homogetization). After grinding, the initial mixture was placed in an oven until the ethanol completely evaporated (T = 80 °C during 30 min). After removing the ethanol, the mixture was sieved through a sieve (mesh 0.04 mm). At the second stage, a mixture of HF/carbon was mixed with a PVDF in specified proportions. Mixing was also carried out by grinding in an agate mortar (without adding ethanol), followed by sieving through sieves (mesh 0.04 mm). After the process of mechanical mixing in a stoichiometric ratio, the mixture was placed in a mold, and the composite samples were thermally pressed (compaction process using external pressure at constant heating). Pressing conditions: pressure 5 MPa; temperature 250 °C; pressing time 20 min. For analysis of the carbon nanomaterials impact we produced 3 samples: 1. HF + PVDF (without carbon nanomaterials); 2. HF + PVDF + CNT and 3. HF + PVDF + CNF.

### 2.2. Characterization Methods

The crystal structure and phase analysis were investigated by XRD in Cu-Kα radiation *λ* = 1.54 Å (Empyrean, PANalytical). The microstructure was studied for all the prepared samples using a scanning electron microscope (SEM) Zeiss EVO 10 (Zeiss, Jena, Germany) with a microanalysis system AZtecLive Advanced with Ultim Max 40 (Oxford Instruments, High Wycombe, UK). The magnetic properties, such as field dependences of the specific magnetization, were investigated by a high-level liquid helium-free measuring system (Cryogenic Ltd., London, UK) for all the prepared samples up to 2 T at room temperature (T = 300 K). Electrodynamic properties as such as frequency dependences of the electrical conductivity were measured on an LRC-meter using the two-probe method (LRC-meter E7-20, Minsk, Belarus).

## 3. Results and Discussion

### 3.1. Structural Parameters

Figure 1 demonstrates XRD patterns of the initial components: organic polymer matrix (PVDF), Al-substituted BaFe_11.7_Al_0.3_O_19_-HF, CNT and CNF. The investigations were performed in the range 15 < 2θ < 80 deg.

X-ray diffraction patterns for PVDF have several diffuse diffraction peaks, which is a common situation for polymers. PVDF has several polymorphous phases, titled α-, β-, and γ-phases. All phases are characterized by the intensive peal near 20 deg. PVDF in β-phase has an intensive peak near 20.4 deg. and PVDF in the γ -phase has a peak near 18 deg. [30,31]. From the XRD patterns we can conclude that PVDF in this case is coexist in a mixed-phase state (coexisting of the β- and γ-phases), which is clearly seen in Figure 2 (XRD patterns for PVDF-based composites) with well-defined diffraction maximums at ~18 and 20.5 deg. On the XRD pattern for HF, we can observe main diffraction peaks from atomic planes (006) at 23.2 deg., (110) at 30.8 deg., (107) at 32.2 deg., (114) at 34.1 deg., (203) at 37.1 deg., (205) at 40.3 deg., (206) at 55.1 deg., (217) at 56.7 deg. and (2011) at. 63.1 deg. that correlates well with classical space group for M-type hexaferrite (magnetoplumbite structure)—P6_3_/mmc [26] with two molecules (Z = 2) per unit cell (Card number 00-007-0276 in ICDD). This confirmed single-phase state of the HF without any impurities. For CNT CNF, some diffraction peaks were observed, which is common for carbon nanomaterials. The most intensive peak was for the atomic plane (002) in the range 26.2–26.8 deg. The XRD pattern of the CNF demonstrates the effect of the texture due to the impact of the quasi-two-dimensional structure and the features of the compactization.

Figure 2 demonstrates XRD patterns of the composites based on initial components: HF + PVDF, HF + PVDF + CNT and HF + PVDF + CNF. The investigations were performed in the range 15 < 2θ < 80 deg. It was shown that the main well-defined diffraction peaks belong to the HF (higher level of the crystallinity of the oxide powder and highest concentration in the final composite. In HF + PVDF we can observe low-intensity peaks for γ-PVDF at 18.2 deg. and diffuse peaks for β-PVDF at 20.4 deg. In HF + PVDF + CNT and HF + PVDF + CNF these peaks were not identified. We assume that it can be a result of the carbon nanomaterial’s impact. For carbon-based samples, the most intensive peak for (002) can be observed. For CNF-based samples, the intensity of this peak is higher in comparison with the CNT-based sample. That is logical if we compare the diffraction data with the intensities of the initial components (see Figure 1). We can conclude from the binary and ternary composites that there is not any destruction of the initial components or any chemical interactions between them. We produced a composite that consist of mixed different phases (organic insulator matrix—PDVF; functional magnetic fillers—HF and highly electroconductive additives—CNT/CNF) in the required ratio.

Figure 3 shows SEM images of the initial components of the final composites: HF (Figure 3a); CNT (Figure 3c); CNF (Figure 3d) and their composites: HF + PVDF (Figure 3b); HF + PVDF + CNT (Figure 3e); HF + PVDF + CNF (Figure 3f). The average grain size for HF powder is 400–500 nm with a hexagonal plate shape (Figure 3a). We can note that there is a high agglomeration of the hexaferrite particles. After mixing HF and PVDF in binary composite (Figure 3b), we can conclude that the density of the sample increases. The polymer matrix fills the pores in HF compacted powder. The CNT is a quasi-one-dimensional structure with a diameter of 50–100 nm and the length ranged from 2 to 10 μm (Figure 3c). CNF is a quasi-two-dimensional structure. It is a thin sheet or lamina with a thickness of 50–70 nm and planar dimensions from 2 to 30 μm.

According to Figure 3, it can be concluded that the formation of the binary and ternary composites with a fixed content of HF (85 wt.%) in the PVDF polymer with various types of carbon nanomaterials (5 wt.%) does not in any way affect the microstructure of the hexaferrite powder, while slightly changing the total density of the sample. The CNT fraction is uniformly distributed on the surface of the grains of the magnetic filler (Figure 3e). It should be noted that in the process of mechanical mixing and thermal pressing, there is a slight decrease in the length of CNT due to mechanical action. CNF is also evenly distributed in the composite material (Figure 3f). Agglomeration of hexaferrite particles on the CNF surface is noted. All samples of composite materials have a good uniform distribution of components. SEM data confirmed that we produced composites that consist of separate phases that combine by mechanically mixing without any destruction of the microstructure and with the preservation of its functional properties.

### 3.2. Magnetic and Electrical Properties

The study of the magnetic characteristics of binary HF + PVDF composite (without carbon) and ternary HF + PVDF + CNT and HF + PVDF + CNF composites were performed by the sample vibrating magnetometry. Investigations of the magnetic characteristics were in the form of field dependences of the specific magnetization at T = 300 K in a wide range of external magnetic fields (up to 2 T). Figure 4 shows the field dependences of the specific magnetization of the composites: HF/PVDF and HF + PVDF + CNT in comparison with the HF. Figure 5 shows the field dependences of the specific magnetization of the composites: HF + PVDF and HF + PVDF + CNF in comparison with the HF.

The main contribution in magnetic properties of this type of composites is the influence of the magnetic phase (HF) and the peculiarities of exchange interactions in a ferrimagnet, since PVDF and CNT/CNF are non-magnetic. The source of the magnetic moment in M-type hexaferrite is the spin-orbital ordering of the magnetic moments in Fe^3+^ ions. In HF the iron ions are placed in different oxygen surroundings or coordination. There are one bipyramidal (2b), one tetrahedral (4f_IV_) and three octahedral (2a, 4f_VI_ and 12k) oxygen environments. The total magnetization in HF is the result of the superposition of the two magnetic sublattices (3 oxygen environments with “spin-up-orientation” and 2 oxygen environments with “spin-down-orientation”). Thus, HF has ferrimagnetic ordering (Gorter’s model). Table 1 demonstrates the main magnetic characteristics of the initial HF, binary HF + PVDF and ternary HF + PVDF + CNT and HF + PVDF + CNF composites. As main magnetic characteristics at T = 300 K we discussed Ms—saturation magnetization; Mr—remnant magnetization; Sq.—squareness ratio (ratio Mr/Ms) and Hc—coercivity.

Pure HF is characterized by the highest magnetic characteristics. This is logical due to the fact that there is only HF has ordered magnetic moments. Thus, Ms and Mr values for HF are 58.1 and 30.2 emu/g, respectively. This leads to the maximal Sq. (0.52) and correlates well with standard multidomain magnetic ceramics. Hc value is 0.37 T or 3.7 kOe. M-type hexaferrites are well-known hard magnetic materials with high magnetic energy (high Mr and Hc) that open perspectives for application as permanent magnets. After mixing of the HF with non-magnetic matrix the main magnetic characteristics Ms, Mr and Hc decreased for binary composite on 7, 10 and 11% till 54.2, 27.2 emu/g and 0.33 T, respectively. It can be associated with a decrease of the magnetitic component mass in composite in comparison with pure HF and a weakening of the intensity of the intragranular exchange interactions. Carbon nanomaterial additives decrease magnetization values. It was also expected that the addition of CNT or CNF would lead to a decrease in Ms and Mr in CM. It can be seen from the data in Figure 4 and Figure 5 that the main magnetic characteristics of the HF + PVDF + CNT and HF + PVDF + CNF are lower than those of the HF + PVDF CM. Thus, for the HF + PVDF the values of Ms (54.2 emu/g), Mr (27.2 emu/g) and Hc (0.33 T). It should be noted that the CM sample with CNT, in contrast to the sample with CNF, enters the state of magnetic saturation in fields up to 0.5 T. HF + PVDF + CNF does not saturate even in fields up to 2 T, which may be due to the weakening of the dipole–dipole exchange interaction due to the influence of the type of carbon nanomaterial. On average, the magnetic characteristics for HF + PVDF + CNT are noticeably lower than for HF + PVDF + CNF. Thus, for HF + PVDF + CNT, the values of Ms (42.9 emu/g) Mr (20.7 emu/g) differ from those for HF + PVDF + CNF where Ms (46.7 emu/g) Mr (23.9 emu/g). The influence of the type of nanosized carbon on such CM characteristics as Hc and Sq. (Mr/Ms ratio) should be separately noted. It is shown that when using quasi-one-dimensional nanocarbon (CNT) in the synthesis of CM, the coefficient Sq (0.48) and the value of Hc (0.07 T) are significantly lower than when using quasi-two-dimensional nanocarbon (CNF), where the coefficient Sq (0.51) and the value of Hc (0.33 T) practically corresponds to the values of the HF + PVDF CM. This fact may be due to the influence of the shape of nanocarbon. It is difficult to present an unambiguous explanation at the moment, however, it can be assumed that this is due to the influence of the weakening of the intergranular exchange interaction in CM when using CNF as compared to CNT, as well as the enhancement of exchange interaction and a decrease in the anisotropy fields in HF + PVDF + CNT.

Figure 6 shows the AC-conductivity (σ_AC_) versus frequency at room temperature for initial HF and HF + PVDF CM. It is noted that with increasing frequency from 500 Hz to 1 MHz, the conductivity of HF changed slightly from 2.47 × 10^−8^ S/cm to 8.57 × 10^−8^ S/cm. While the conductivity of HF + PVDF CM in the same frequency range changed almost by 2 orders of magnitude, from 5.31 × 10^−9^ S/cm to 9.36 × 10^−7^ S/cm. Many authors, when discussing the electrical properties of CMs based on hexaferrite and conducting matrices, interpreted the conductivity as increasing with frequency in terms of Koop’s models in the approximation of space charge polarization.

Figure 7 shows the AC-conductivity (σ_AC_) versus frequency at room temperature for HF + PVDF + CNT and HF + PVDF + CNF CMs. A sharp increase in the electrical conductivity values for CMs with the addition of nanosized carbon (by more than 7 orders of magnitude) was noted, which is due to the high concentration of free charge carriers in CNT and CNF and, as a consequence, higher electrical conductivity. AC conductivity is due to both the contribution of the grain boundary component (transport of charge carriers along defective areas of grain boundaries) and the bulk conductivity of the material (which is a combination of capacitances and resistances). In the case of the investigated CM, the bulk conductivity mechanism can be represented as the transport of charge carriers through an inhomogeneous medium consisting of conductive inclusions (nanosized carbon derivatives) separated by a dielectric layer (PVDF and hexaferrite). Such a system has a characteristic recharge time τ ≈ RC. If the electric capacitance at high frequency does not have time to be recharged, this hinders the transport of charge carriers and causes a sharp decrease in the electrical conductivity value. According to Figure 7, it is obvious that, depending on the type of nanosized carbon (CNT or CNF), the electrical characteristics of ferrite/polymer CM based on HF + PVDF differ significantly. Thus, for HF + PVDF + CNT CM, the electrical conductivity was more than two times higher than for HF + PVDF + CNF. It was noted that in the frequency range from 500 Hz to 10^4^ Hz, the conductivity changed insignificantly, both for HF + PVDF + CNT (1.3–1.11 S/cm) and for HF + PVDF + CNF (0.65–0.63 S/cm).

For granular composite systems, a sharp decrease in electrical conductivity with increasing frequency can also be due to the inhomogeneity of the distribution of the contact resistance value between the particles of the carbon filler. Thus, a larger number of contacts between the conducting component (CNT or CNF) in a fixed CM volume leads to an increase in electrical conductivity. This can explain the higher value of electrical conductivity for HF + PVDF + CNT compared to HF + PVDF + CNF (for CNT, due to the size factor, the number of contacts of the conductive component per unit volume is higher, and this is isotropic in contrast to CNF). Furthermore, the difference in conductivity may be due to the volume inhomogeneity of the distribution of the carbon filler in the polymer matrix due to the insufficient degree of dispersion of carbon fillers or due to their agglomeration (the surface energy of quasi-two-dimensional objects of the CNF type is much higher than for quasi-one-dimensional objects of the CNT type, which can cause a greater tendency to agglomeration), the shape of the filler particles, the nature of the interaction of fillers with the matrix, the presence of contact phenomena at the particle-particle interface. With an increase in frequency above 10^4^ Hz, the conductivity for HF + PVDF + CNT and HF + PVDF + CNF sharply decreases to 0.67 and 7 × 10^−3^ S/cm, respectively. For a more detailed analysis and formation of a model that explains the nature of the influence of the shape of nanosized carbon in the framework of the classical percolation theory, an analysis of the behavior of concentration dependences is required.

## 4. Conclusions

To establish the influence of the type of carbon nanomaterials (quasi-one-dimensional CNT and quasi-two-dimensional CNF) on magnetic and electrodynamic parameters we produced ternary composites based on Al-substituted HF (BaFe_11.7_Al_0.3_O_19_) in a polymer matrix (PVDF). The ratio between HF and PVDF in HF + PVDF composite was fixed (85 wt% HF and 15 wt% PVDF). The concentration of CNT and CNF in CM was fixed (5 wt% from total HF + PVDF weight). Composites were prepared by mechanical mixing of the initial powders with the thermal pressing of the mixture. XRD patterns demonstrate that CM contains single-phase HF, γ- and β-PVDF and carbon nanomaterials. Thus, we produced composites that consist of mixed different phases (organic insulator matrix—PDVF; functional magnetic fillers—HF and highly electroconductive additives—CNT/CNF) in the required ratio. There were not any chemical interactions between the initial components. All samples consist of separate phases that combine by mechanically mixing without any destruction of the microstructure and with the preservation of its functional properties. SEM images demonstrate to us that all samples of composite materials have a good uniform distribution of components. Investigations of the magnetic characteristics were in the form of field dependences of the specific magnetization at T = 300 K in a wide range of external magnetic fields (up to 2 T). It was demonstrated that the main contribution in magnetic properties of this type of composite belongs to the magnetic phase (HF) and the peculiarities of exchange interactions in a ferrimagnet, since PVDF and CNT/CNF are non-magnetic. We demonstrated the features of the main magnetic characteristics for investigated composited and discussed the differences for CNT and CNF-based CM. This fact may be due to the influence of the shape of the nanocarbon. It is difficult to present an unambiguous explanation at the moment, however, it can be assumed that this is due to the influence of the weakening of the intergranular exchange interaction in CM when using CNF as compared to CNT, as well as the enhancement of exchange interaction and a decrease in the anisotropy fields in HF + PVDF + CNT. The rapid increase of the electrical AC-conductivity was observed for HF + PVDF + CNT and HF + PVDF + CNF in comparison with the binary HF + PVDF composite. The reason for the differences in electrodynamic properties was discussed for binary and ternary composites. These results open broad perspectives for the development and analysis of the ternary nanocomposites based on different phases (organic insulator matrix; functional magnetic fillers and highly electroconductive additives). First, the obtained results will help us to predict the impact of the shape of the nanosized carbon materials on the physical properties of the composites. Second, the developed materials can be used in the field of high-frequency applications such as antenna technology (5G or 6G range) and electromagnetic compatibility. In further investigations, we will demonstrate the correlation of microwave properties vs. nanocarbon.

## Figures and Tables

**Figure 1 nanomaterials-12-00868-f001:**
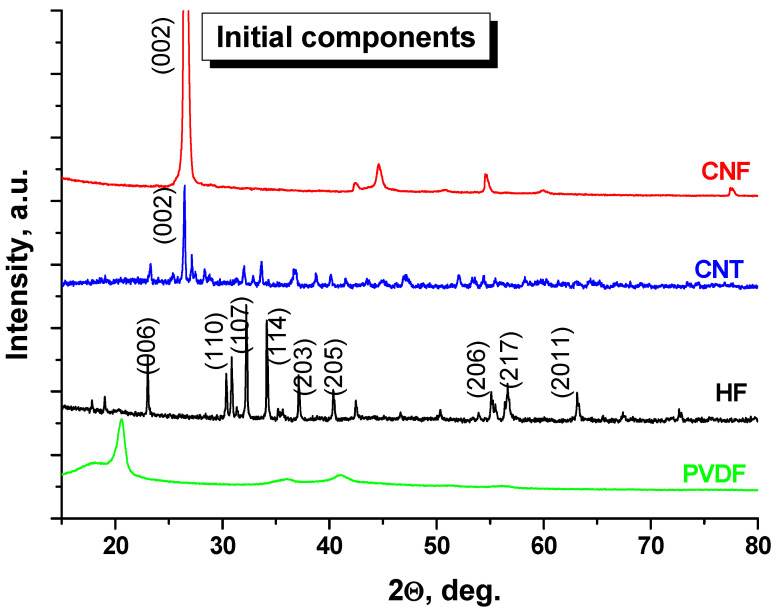
XRD patterns of the initial components: PVDF; HF; CNT and CNF.

**Figure 2 nanomaterials-12-00868-f002:**
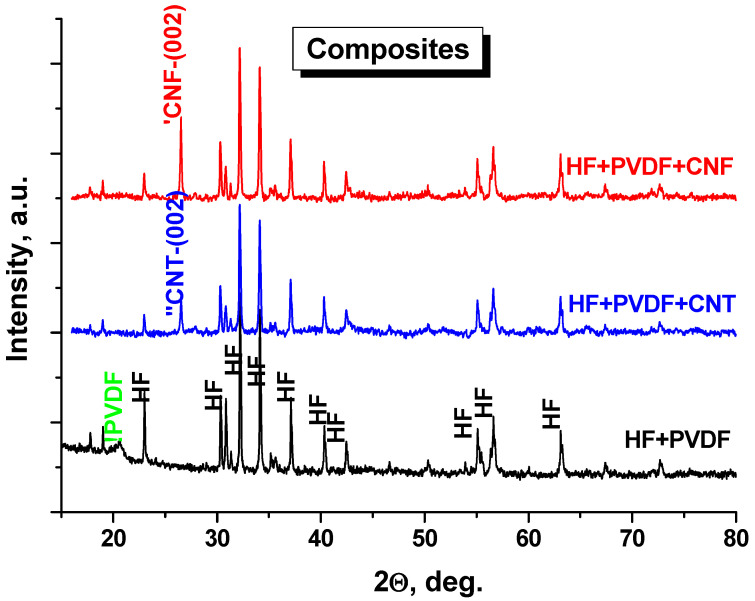
XRD patterns of the composites based on initial components: HF + PVDF; HF + PDF + CNT and HF + PVDF + CNF.

**Figure 3 nanomaterials-12-00868-f003:**
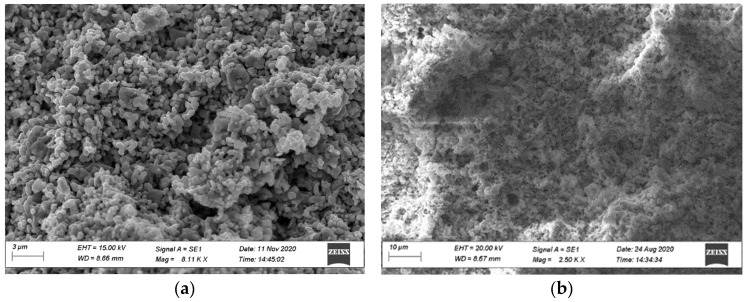
SEM images of the initial components: HF (**a**); HF + PVDF (**b**); CNT (**c**); CNF (**d**); and composites: HF + PDF + CNT (**e**) and HF + PVDF + CNF (**f**).

**Figure 4 nanomaterials-12-00868-f004:**
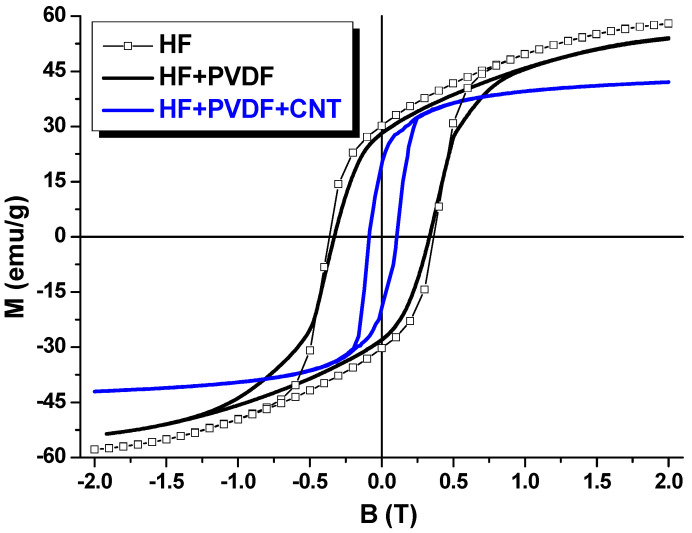
Magnetic field dependences of the specific magnetization of the HF; HF + PVDF and HF + PDF + CNT.

**Figure 5 nanomaterials-12-00868-f005:**
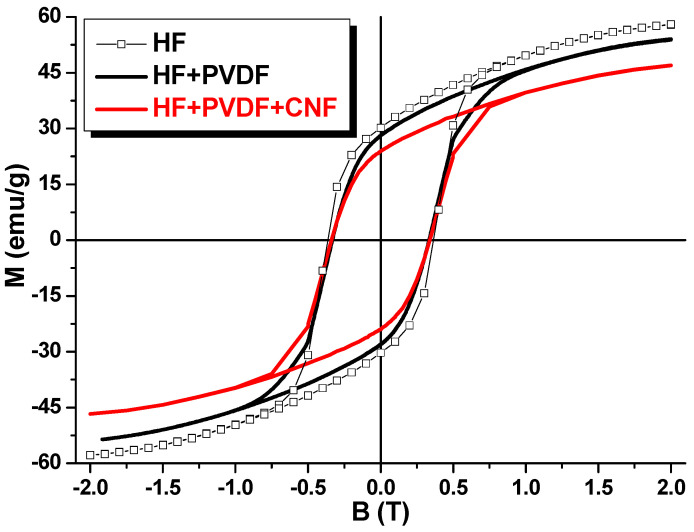
Magnetic field dependences of the specific magnetization of the HF; HF + PVDF and HF + PDF + CNF.

**Figure 6 nanomaterials-12-00868-f006:**
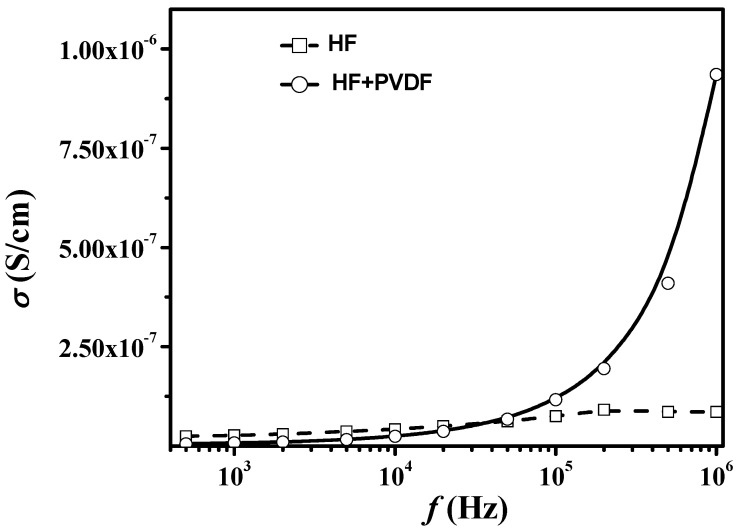
Frequency dependences of the electrical conductivity of the HF and HF + PVDF.

**Figure 7 nanomaterials-12-00868-f007:**
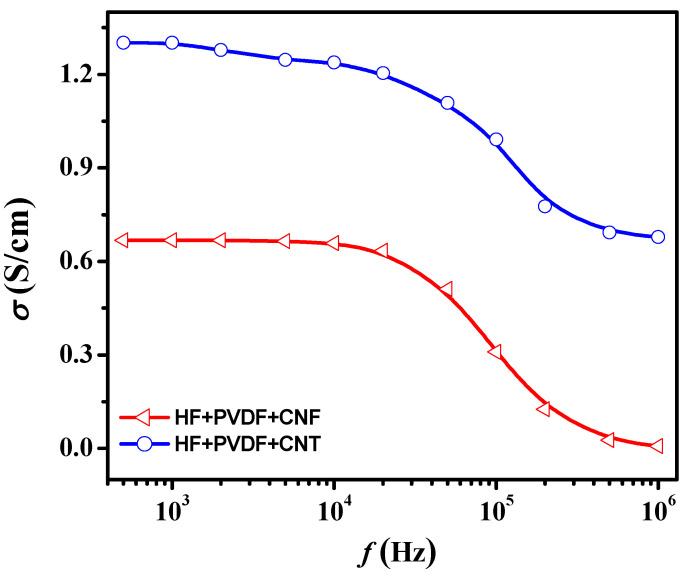
Frequency dependences of the electrical conductivity of the HF + PVDF + CNT and HF + PVDF + CNF.

**Table 1 nanomaterials-12-00868-t001:** Main Magnetic characteristics of the initial HF, binary HF + PVDF and ternary HF + PVDF + CNT and HF + PVDF + CNF composites: Ms—saturation magnetization; Mr—remnant magnetization; Sq.—squareness ratio (ratio Mr/Ms) and Hc—coercivity.

Magnetic Characteristics	Al-HF [20]	In-HF [19]	Al-HF [28]	HF	HF + PVDF	HF + PVDF + CNT	HF + PVDF + CNF
Ms, emu/g	45.33	51.01	65.1	58.1	54.2	42.9	46.7
Mr, emu/g	4.6	8.29	34.2	30.2	27.2	20.7	23.9
Sq. (Mr/Ms)	0.1	0.16	0.52	0.52	0.50	0.48	0.51
Hc, kOe	0.1	0.8	3.9	3.7	3.3	0.7	3.3

## Data Availability

The data presented in this study are available on request from the corresponding authors.

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
