# Peer review of "Impact of the Nanocarbon on Magnetic and Electrodynamic Properties of the Ferrite/Polymer Composites"

_nanomaterials, 2022, doi:10.3390/nano12050868_

Round 1

Reviewer 1 Report

The manuscript entitled “Impact of the Nanocarbon on Magnetic and Electrodynamic Characteristics of the Ferrite/Polymer composites” is devoted to preparation and investigation of the ferrite-based composites, prepared by mixing of the hexagonal ferrite and PVDF with nanocarbon. The topic of this paper is critically actual especially for the areas of functional magnetic composites. The paper describes how the shape of the nanocarbon (tubes or flakes) impact on magnetic and electrical properties of the three-phase composites. Authors compared the result of the three-phase composites with sample without carbon. Nevertheless, there are several points before the paper can be published. Manuscript can be accepted after a minor revision with the following comments.  

1.    I think that the title of the paper doesn’t cover the topic of research broadly enough. It should be re-written. I propose replace “Characteristics” by the “Properties”.
2.    Abstract seems poor. Please add most important results and highlight them in abstract.
3.    The motivation of the research objects must be highlighted in Introduction part. Why authors used Al-substituted hexagonal ferrites? Please add some information about the Al-substitution for hexaferrites.
4.    Section 2.1 contain non-uniform details. It the text authors titled composited using “+” like HF+PVDF. But in section 2.1 I found “/” like HF/PVDF/CNT”. Please revise this. The same for section 3.1 (line 136) where authors used “HF+PVDF-CNT”. Please check this carefully.
5.    Please check is Space Group (P63/mmc) is correct? I feel that “3” must be as lower index.
6.    The description about experimental equipment is absent in experimental part as well as SEM technique. 
7.     Very interesting explanation of the differences of the magnetic properties for CNT- and CNF-based composites. Is any explanation of the differences between HF and HF+PVDF?
8.    Table 1 – I feel that for “Hc” will better use “Oe” or “A/m”. Units “T” used for magnetic induction.
9.    The Conclusion part is too short, please improve it. The practical recommendation should be added to the text of conclusion.
10.    There are some insufficient typos and English mistakes in the text. Please check this and provide revised version.

Author Response

Referee #1 Comments and Suggestions for Authors

The manuscript entitled “Impact of the Nanocarbon on Magnetic and Electrodynamic Characteristics of the Ferrite/Polymer composites” is devoted to preparation and investigation of the ferrite-based composites, prepared by mixing of the hexagonal ferrite and PVDF with nanocarbon. The topic of this paper is critically actual especially for the areas of functional magnetic composites. The paper describes how the shape of the nanocarbon (tubes or flakes) impact on magnetic and electrical properties of the three-phase composites. Authors compared the result of the three-phase composites with sample without carbon. Nevertheless, there are several points before the paper can be published. Manuscript can be accepted after a minor revision with the following comments.  

Comment 1.    I think that the title of the paper doesn’t cover the topic of research broadly enough. It should be re-written. I propose replace “Characteristics” by the “Properties”.

Response: Dear Reviewer, thank you for your comment. We changed the title as was proposed.

Comment 2.    Abstract seems poor. Please add most important results and highlight them in abstract.

Response: Dear Reviewer, thank you for your comment. We did this as requested. Please find revised abstract in manuscript.

Comment 3.    The motivation of the research objects must be highlighted in Introduction part. Why authors used Al-substituted hexagonal ferrites? Please add some information about the Al-substitution for hexaferrites.

Response: Dear Reviewer, thank you for your comment. We added part that described the motivation of the Al-substitution. Please find in revised manuscript (Introduction):

“The most interesting case for controlling magnetic, electrical and electrodynamic properties is the substitution of the Fe3+ by the Al3+ ions [20-23]. Al3+ ions are diamagnetic ions and have different ionic radius and electronic configuration that results in the changes in magnetic and electrical properties of the substituted HF.”

Comment 4.    Section 2.1 contain non-uniform details. It the text authors titled composited using “+” like HF+PVDF. But in section 2.1 I found “/” like HF/PVDF/CNT”. Please revise this. The same for section 3.1 (line 136) where authors used “HF+PVDF-CNT”. Please check this carefully.

Response: Dear Reviewer, thank you for your comment. We revised this. Sorry us, it was typo.

Comment 5.    Please check is Space Group (P63/mmc) is correct? I feel that “3” must be as lower index.

Response: Dear Reviewer, thank you for your comment. Yes, You are right. We revised this.

Comment 6.    The description about experimental equipment is absent in experimental part as well as SEM technique. 

Response: Dear Reviewer, thank you for your comment. We added part anout the experimental equipment (SEM also). Now it seems:

“…Microstructure was studied for all the prepared samples using a scanning electron microscope (SEM) Zeiss EVO 10 (Zeiss, Germany) with a microanalysis system AZtecLive Advanced with Ultim Max 40 (Oxford Instruments, UK). …

Comment 7.     Very interesting explanation of the differences of the magnetic properties for CNT- and CNF-based composites. Is any explanation of the differences between HF and HF+PVDF?

Response: Dear Reviewer, thank you for your comment. Yes, We added explanation about the origin of the differences between magnetic properties of the pure HF and HF+PVDF composites. Please find bellow the brief explanation:

“The main contribution in magnetic properties of these type composites is due to the influence of the magnetic phase (HF) and the peculiarities of exchange interactions in a ferrimagnet, since PVDF and CNT/CNF are non-magnetic. Pure HF is characterized by the highest magnetic characteristics. This is logical due to there is only HF has ordered magnetic moments. Thus, Ms and Mr values for HF are 58.1 and 30.2 emu/g respectively. This leads to the maximal Sq. (0.52) and correlates well with standard multidomain magnetic ceramics. Hc value is 0.37 T or 3.7 kOe. M-type hexaferrites are well known hard magnetic materials with high magnetic energy (high Mr and Hc) that open perspectives for application as permanent magnets. After mixing of the HF with non-magnetic matrix the main magnetic characteristics Ms, Mr and Hc decreased for binary composite on 7, 10 and 11% till 54.2, 27.2 emu/g and 0.33 T respectively. It can be associated with decrease of the magnetitic component mass in composite in comparison with pure HF and weakening in the intensity of the intragranular exchange interactions.”

Thus, there are 2 main reasons: i. decrease of the magnetitic component mass in composite and ii. weakening in the intensity of the intragranular exchange interactions.

Comment 8.    Table 1 – I feel that for “Hc” will better use “Oe” or “A/m”. Units “T” used for magnetic induction.

Response: Dear Reviewer, thank you for your comment. We revised this. We used “kOe” units.

Comment 9.    The Conclusion part is too short, please improve it. The practical recommendation should be added to the text of conclusion.

Response: Dear Reviewer, thank you for your comment. We did this as requested. Please find in revised conclusions new information about the practical recommendation:

“These results open broad perspectives for development and analysis of the ternary nanocomposites based on different phases (organic insulator matrix; functional magnetic fillers and highly electroconductive additives). First, obtained results will help us to predict the impact of the shape of the nanosized carbon materials on physical properties of the composites. Second, the developed materials can be used in the field of the high-frequency applications such as antenna technology (5G or 6G range) and for electromagnetic compatibility. In further investigations, we will demonstrate the correlation of the microwave properties vs. nanocarbon.”

Comment 10.    There are some insufficient typos and English mistakes in the text. Please check this and provide revised version.

Response: Dear Reviewer, thank you for your comment. We did brutal revision and language editing. All changes were highlighted in yellow.

General comment from authors:

Many thanks for your comments. All comments were useful for us and we hope it helped us do our manuscript better.

Reviewer 2 Report

Title: Impact of the Nano carbon on Magnetic and Electrodynamic Characteristics of the Ferrite/Polymer composites Binary and ternary composites based on M-type hex ferrite (HF), polymer matrix (PVDF) and carbon nanomaterials (quasi-one-dimensional carbon nanotubes – CNT and quasi two-dimensional carbon Nano flakes – CNF) were prepared and investigated for establishing of the impact of the different Nano sized carbon on magnetic and electrodynamic properties. The novelty of this work, as well as its interesting results, is clearly described in the article. Therefore, the referee would like to recommend this work to minor revision and to be published after careful consideration according to the comments below: 1. The main novelty of this paper should be clarified and should be added to the last paragraph of the introduction. 2. Please add more current papers in the literature and improve the introduction section. Some interesting papers related to the topic of this manuscript could be: Thermo-electro-mechanical size-dependent postbuckling response of axially loaded piezoelectric shear deformable nanoshells via nonlocal elasticity theory, Microsystem Technologies 23 (10), 5105-5119 3. The writing of this paper needs to be improved and polished. Some clumsy and neglectful expressions can be found.

Author Response

Referee #2 Comments and Suggestions for Authors

Title: Impact of the Nano carbon on Magnetic and Electrodynamic Characteristics of the Ferrite/Polymer composites Binary and ternary composites based on M-type hex ferrite (HF), polymer matrix (PVDF) and carbon nanomaterials (quasi-one-dimensional carbon nanotubes – CNT and quasi two-dimensional carbon Nano flakes – CNF) were prepared and investigated for establishing of the impact of the different Nano sized carbon on magnetic and electrodynamic properties. The novelty of this work, as well as its interesting results, is clearly described in the article. Therefore, the referee would like to recommend this work to minor revision and to be published after careful consideration according to the comments below: 1. The main novelty of this paper should be clarified and should be added to the last paragraph of the introduction. 2. Please add more current papers in the literature and improve the introduction section. Some interesting papers related to the topic of this manuscript could be: Thermo-electro-mechanical size-dependent postbuckling response of axially loaded piezoelectric shear deformable nanoshells via nonlocal elasticity theory, Microsystem Technologies 23 (10), 5105-5119 3. The writing of this paper needs to be improved and polished. Some clumsy and neglectful expressions can be found.

Response: Dear Reviewer, thank you for your comment. We did all corrections as requested.

  1. We added novelty in the last paragraph of the Introduction. Now it seems like:

We need to note that it was first time when we demonstrated and analyzed the critical impact of the type of the nanocarbon (first of all the shape and surface activity) on magnetic and electrodynamic properties.

  1. We added relevant literature to the introduction. In addition, we carefully checked proposed paper and added to reference list (please check [24]).
  2. Dear Reviewer, thank you for your comment. We did brutal revision and language editing. All changes were highlighted in yellow.

General comment from authors:

Many thanks for your comments. All comments were useful for us and we hope it helped us do our manuscript better.

Reviewer 3 Report

The manuscript by Trukhanov et al. “Impact of the Nanocarbon on Magnetic and Electrodynamic Characteristics of the Ferrite/Polymer composites” requires revision to address major concerns.

Comments.

  1. Lines 32-34, please elaborate in detail with particular applications in the citations.
  2. The novelty and significance of the present study should be provided at the end of the Introduction.
  3. The application of materials is missing. Please add one real application of synthesized material with corresponding controls to highlight its suitability.
  4. In addition, the authors should provide the literature search (comparison Table) to justify the significance of synthesizing a superior material.
  5. The authors should add high-resolution SEM images showing individual particles (not aggregated particles). Please add particles distribution data.
  6. Please add elemental mapping data present in the various materials for more justification.
  7. Overall, the discussion section can be improved.

Author Response

Referee #3 Comments and Suggestions for Authors

The manuscript by Trukhanov et al. “Impact of the Nanocarbon on Magnetic and Electrodynamic Characteristics of the Ferrite/Polymer composites” requires revision to address major concerns.

Comments.

Comment 1: Lines 32-34, please elaborate in detail with particular applications in the citations.

Response: Dear Reviewer, thank you for your comment. We discussed in detail the particular applications. Please find this in revised version:

“Today, the development of the functional composites based on magnetic (such as ceramic/ceramic or hard/soft composites [1-3] and ceramic/polymer composites [4, 5]) and carbon (nanotubes, graphite and nanoplatelet) [6-10] nanomaterials attracts great attention due to the fundamental significance and practical importance. The fundamental significance is associated with the violation of the principle of additivity. From the practical point of view these composites can be used for various application in functional devices as active media [2, 3, 5]. These materials can be used in the field of the high-frequency applications such as electromagnetic absorbers [1-3], antenna technology [6] and for electromagnetic compatibility [10].”

Comment 2: The novelty and significance of the present study should be provided at the end of the Introduction.

Response: Dear Reviewer, thank you for your comment. We added novelty in the last paragraph of the Introduction. Now it seems like:

We need to note that it was first time when we demonstrated and analyzed the critical impact of the type of the nanocarbon (first of all the shape and surface activity) on magnetic and electrodynamic properties.

Comment 3: The application of materials is missing. Please add one real application of synthesized material with corresponding controls to highlight its suitability.

Response: Dear Reviewer, thank you for your comment. Please find in revised conclusions new information about the real practical applications:

“These results open broad perspectives for development and analysis of the ternary nanocomposites based on different phases (organic insulator matrix; functional magnetic fillers and highly electroconductive additives). First, obtained results will help us to predict the impact of the shape of the nanosized carbon materials on physical properties of the composites. Second, the developed materials can be used in the field of the high-frequency applications such as antenna technology (5G or 6G range) and for electromagnetic compatibility. In further investigations, we will demonstrate the correlation of the microwave properties vs. nanocarbon.”

Comment 4: In addition, the authors should provide the literature search (comparison Table) to justify the significance of synthesizing a superior material.

Response: Dear Reviewer, thank you for your comment. Sorry us, but we didn’t find any composites with the similar composition (we mean especially Al-substituted hexaferrite in amount 85 wt.% + 15 wt.% PVDF and appropriate quality carbon nanomaterials with the same concentration 5wt. %). Thus we think that it will no correct to compare the magnetic properties of the composites with the different compositions. We added new columns for comparison of the Al-susbtituted HF (from [20, 28]) and In-susbtitued HF (from [19]) with the same concentration of the substituents. Hope on your understanding and support.

Comment 5: The authors should add high-resolution SEM images showing individual particles (not aggregated particles). Please add particles distribution data.

Response: Dear Reviewer, thank you for your comment. Please let us to explain. HF has a high resistivity and when we add PVDF (with ferroelectric behavior) it’s hardly realize the high resolution images doe to polarization of the electron beam. Sorry us, but we can’t provide at this stage high resolution SEM-images and as result we can’t calculate particle size distribution. The microstructure analysis is not our main goal of this paper. Hope on your understanding and support.

Comment 6: Please add elemental mapping data present in the various materials for more justification.

Response: Dear Reviewer, thank you for your comment. Please let us to explain. When we analyze composites based on organic matrix (PVDF) and carbon additives – we obtain the complex task for elemental analysis – there are a lot of C peaks on EDX and as a result we will obtain incorrect elemental mapping – there is a lot of C and low level of the Ba and Al on maps. It will be no informative. Sorry us, but we think it will no more justification if will will provide elemental mapping. The element distribution is not our main goal of this paper. Hope on your understanding and support.

Comment 7: Overall, the discussion section can be improved.

Response: Dear Reviewer, thank you for your comment. We improved the discussion section as requested. Please find this in revised version. Please let us share by the results in Nanomaterials. Hope on your help and support.

General comment from authors:

Many thanks for your comments. All comments were useful for us and we hope it helped us do our manuscript better.

Round 2

Reviewer 3 Report

The authors have partially revised the manuscript. It may be accepted.